# The role of comorbidities in the associations between air pollution and Alzheimer's disease: A national cohort study in the American Medicare population

**Yanling Deng**[ID]*, **Yang Liu, Hua Hao, Ke Xu**[ID]**, Qiao Zhu, Haomin Li**[ID]**, Tszshan Ma, Kyle Steenland**

Gangarosa Department of Environmental Health, Rollins School of Public Health, Emory University, Atlanta, Georgia, United States of America

* yanling.deng@emory.edu

## Abstract

### Background

Air pollution and several common comorbidities—such as hypertension, stroke, and depression—are established risk factors for Alzheimer's disease (AD). However, whether these comorbidities mediate or amplify the effects of fine particulate matter ($PM_{2.5}$) on AD remains unclear. We aimed to investigate whether these conditions modify or mediate the association between $PM_{2.5}$ exposure and incident AD.

### Methods and findings

We conducted a nationwide cohort study including 27.8 million US Medicare beneficiaries aged 65 years and older from 2000 to 2018. Exposure to $PM_{2.5}$ was assessed using high-resolution air pollution datasets. Cox proportional hazards models were applied to estimate the associations between exposure to $PM_{2.5}$, incident AD, and comorbidities. The potential for comorbidities to modify and mediate the association between $PM_{2.5}$ and AD was evaluated by stratified analyses and mediation analysis. We identified approximately 3.0 million incident AD cases. $PM_{2.5}$ exposure (5-year moving average prior to AD onset) was associated with increased risk of AD in the overall population (hazard ratio [HR]) per interquartile range [IQR, 3.8 μg/m³] increase: 1.085 (95% CI: 1.078, 1.091]. This association was slightly stronger in individuals with stroke (HR per IQR increase: 1.105; 95% CI: 1.096, 1.114), but there was little effect modification for hypertension and depression. $PM_{2.5}$ exposure was also significantly associated with higher risks of hypertension, depression, and stroke, all of which were also linked to increased AD risk. However, mediation effects were minimal, with 1.6% of the association between $PM_{2.5}$ and incident AD mediated by hypertension, 4.2% by stroke, and 2.1% by depression. Study limitations include

**Data availability statement:** The authors are not permitted to share the third-party raw data used in the analyses because the data contain protected health information of Medicare beneficiaries and are subject to the data use agreement with the Centers for Medicare & Medicaid Services (CMS). Individual researchers may request access to the data by submitting an application through the CMS Research Data Assistance Center (ResDAC) (https://www.resdac.org/).

**Funding:** This work was supported by the National Institutes of Health (https://www.nih.gov/) (R01 AG074357 to KS and R01 ES034175 to YL). The funders had no role in study design, data collection and analysis, decision to publish, or preparation of the manuscript.

**Competing interests:** The authors have declared that no competing interests exist.

**Abbreviations:** Aβ, amyloid-β; AD, Alzheimer's disease; ADRD, AD and related dementias; APOE4, Apolipoprotein E allele 4; BMI, body mass index; CCW, Chronic Conditions Warehouse; CI, confidence interval; CVD, cardiovascular diseases; DE, direct effect; FFS, fee-for-service; GEE, generalized estimating equations; HPC, high-performance computing; HR, hazard ratio; ICD, International Classification of Diseases; IE, indirect effect; IQR, interquartile range; PM$_{2.5}$, fine particulate matter; SES, socioeconomic status; TDP-43, TAR DNA/RNA-binding protein 43; TE, total effect; U.S., United States.

use of administrative claims data and potential exposure misclassification from area-level PM$_{2.5}$ estimates.

## Conclusions

Our findings suggest that PM$_{2.5}$ exposure was associated with increased AD risk, primarily through direct rather than comorbidity-mediated pathways. Stroke may modestly increase susceptibility. These findings highlight the need for air quality interventions as part of dementia prevention strategies in aging populations, especially those facing overlapping environmental and clinical vulnerabilities.

---

## Author summary
### Why was this study done?

- Alzheimer's disease (AD) is the most common form of dementia and a growing public health challenge, especially in aging populations.

- Exposure to air pollution, particularly fine particulate matter (PM$_{2.5}$), has been linked to an increased risk of AD, likely through pathways such as neuroinflammation, oxidative stress, and vascular injury, though these mechanisms remain complex and not fully understood.

- Common chronic health conditions—such as hypertension, stroke, and depression—are also linked to AD, and may help explain or amplify the effects of air pollution on brain health.

- We aimed to determine whether these conditions act as mediators (air pollution leads to comorbidities, which in turn lead to AD) or modifiers (the effect of air pollution is stronger in the presence of comorbidities) in the relationship between PM$_{2.5}$ exposure and AD risk.

### What did the researchers do and find?

- We conducted a national cohort study to jointly evaluate both mediation and effect modification by hypertension, stroke, and depression in the association between 5-year time-varying average PM$_{2.5}$ exposure and incident AD over 27.8 million US Medicare beneficiaries aged 65 years and older from 2000 to 2018.

- We found that 5-year average PM$_{2.5}$ exposure prior to AD onset was associated with an increased risk of AD.

- This association was slightly stronger in individuals who had experienced a stroke, suggesting they may be more vulnerable.

- Although all three comorbidities, including hypertension, stroke, and depression, were associated with both PM$_{2.5}$ exposure and AD risk, the effect of air pollution was largely independent of the presence of comorbidities, which did not act as mediators.

### What do these findings mean?

- Air pollution may contribute to AD mostly through direct pathways rather than through other chronic health conditions.

- Individuals with a history of stroke may be especially susceptible to the harmful effects of air pollution on brain health.

- Improving air quality could be an important way to prevent dementia and protect older adults—particularly those with existing health risks.

- The main limitation is that disease diagnoses and air pollution exposure were based on healthcare records and residential areas, respectively.

## Introduction

Alzheimer's disease (AD), characterized by a gradual decline in memory, cognition, and behavior, is a progressive neurodegenerative disorder and the most common type of dementia [1]. In 2019, over 5 million people in the United States (U.S.) and approximately 57 million worldwide were affected by AD and related dementias (ADRD) [2]. This number is expected to double in the U.S. and triple globally by 2050 [2]. As there are currently no disease-modifying treatments for ADRD, identifying modifiable risk and protective factors is a critical priority for both clinical practice and public health [3].

Emerging evidence indicates that exposure to air pollution—particularly fine particulate matter ($PM_{2.5}$)—is a novel modifiable risk factor associated with the development of AD and dementia [4–6]. Although the exact mechanisms linking $PM_{2.5}$ to ADRD are not fully understood, research suggests that neuroinflammation, oxidative stress, and chronic comorbidities may play significant roles in this relationship [7–9]. Moreover, neuropathological studies indicate that exposure to $PM_{2.5}$, ultrafine particles, and industrial nanoparticles is associated with early deposition of amyloid-β (Aβ), hyperphosphorylated tau tangles, α-synuclein, and TAR DNA/RNA-binding protein 43 (TDP-43) in children and young adults [10–12]. These pathological features closely resemble those observed in older adults and support a life-course continuum of pollution-related neurodegeneration that may culminate in late-life clinical ADRD [13]. Long-term exposure to $PM_{2.5}$ has also been shown to increase the risk of chronic diseases such as cardiovascular conditions [14,15], hypertension [16], and mental health disorders [17,18]. Furthermore, growing evidence, including our previous analysis of the same Medicare data, suggests that comorbidities—particularly hypertension, stroke, and depression—are strongly associated with higher rates of AD and ADRD [19,20]. It is plausible to hypothesize that certain comorbidities may contribute to the association of $PM_{2.5}$ with AD and ADRD.

Identifying modifiable intermediates in causal pathways is crucial for public health research, as they present key opportunities for effective population-level interventions. To date, few studies have investigated the role of comorbidities as intermediaries between $PM_{2.5}$ and dementia incidence, resulting in inconsistent results [21–24]. Some studies suggest that cardiovascular diseases (CVD), especially stroke, mediated this relationship [21,22], while others find no evidence of mediation through hypertension or stroke [23,24]. However, these studies are constrained by small sample sizes and lack adequate mediation analyses that account for interactions between exposure and potential mediators. Additionally, much existing research has focused on dementia as a composite outcome, whereas our study specifically examined clinically diagnosed AD based on validated Medicare algorithms, although some degree of diagnostic overlap with other neurodegenerative dementias is certainly possible [11]. To our knowledge, no study has simultaneously evaluated the roles of incident stroke, hypertension, and depression as both mediators and modifiers of the associations between exposure to $PM_{2.5}$ and incident AD.

To address this gap, we conducted a nationwide cohort study of more than 27 million older adults in the U.S. to evaluate the roles of three common comorbidities—hypertension, stroke, and depression—as potential mediators and effect modifiers of the association between $PM_{2.5}$ exposure and incident AD, while controlling for multiple potential confounders. By leveraging a large, population-based sample and advanced analytical methods, this study aims to provide robust

evidence on the interplay between PM$_{2.5}$, comorbidities, and AD risk, with implications for both clinical practice and public health policy.

## Materials and methods

### Study design and population

We utilized data from two nationwide, privacy-protected databases from the Centers for Medicare and Medicaid Services: the Medicare denominator file and the Medicare Chronic Conditions Warehouse (CCW) numerators file to establish open cohorts, where people could enter and leave at any time from 2000 to 2018. The Medicare denominator file provided annual enrollment records, including beneficiaries' Medicaid insurance status (reflecting socioeconomic status [SES]), age, sex, race, ZIP code, date of death (if applicable), and months enrolled in Part A, Part B, or HMO. Age, Medicaid eligibility, and ZIP code information are updated annually. This research received approval from Emory's Institutional Review Board (IRB, #RSCH-2020-55733) and was conducted under a data use agreement with CMS (#STUDY00000316).

Participants were included if they met the following criteria: (1) aged 65 or older; (2) resided in the contiguous U.S. while enrolled in Medicare; (3) enrolled in the Medicare fee-for-service program; and (4) had both Part A (hospital insurance) and Part B (medical insurance) during the follow-up years. Follow-up years were restricted to participants enrolled exclusively and consecutively in the fee-for-service part of Medicare; if not enrolled in fee-for-service (e.g., if enrolled in Medicare Advantage), data on disease incidence were not available. To better capture newly recorded AD diagnoses rather than previously documented cases, we implemented a 5-year "clean period" prior to cohort entry. This approach aims to minimize misclassification of prevalent cases as incident based on Medicare claims, but does not imply that participants were biologically free of early disease during this period. Given the long preclinical phase of AD and the delay between symptom onset and diagnosis in administrative data, we considered a 5-year period to be a reasonable and widely used approach for reducing the likelihood of including previously diagnosed AD cases in Medicare-based studies of AD or dementia [20,25,26]. Consequently, participants entered the cohort on January 1 of the year following this clean period and were followed until their first AD diagnosis, death, or the end of the follow-up period. The 5-year clean period was excluded from the follow-up time, as participants were not considered at risk for AD during this duration.

### Exposure assessment

High-resolution daily PM$_{2.5}$ concentrations (24-hour averages) across the U.S. were obtained at a spatial resolution of 1 km × 1 km using spatiotemporal ensemble models that incorporated various machine learning algorithms, including gradient boosting, neural networks, and random forests. Detailed methods can be found in prior research [27]. In brief, the ensemble model was calibrated with numerous predictors, such as land-use data, meteorological variables, chemical transport model simulations, satellite measurements, and monitoring data from the Environmental Protection Agency (EPA) Air Quality Systems (AQS). This approach yielded a robust PM$_{2.5}$ prediction model, achieving an average cross-validated coefficient of determination ($R^2$) of 0.89 [27].

Daily PM$_{2.5}$ exposures were aggregated by ZIP code, averaged annually, and linked to each Medicare beneficiary according to their ZIP code of residence and follow-up year. We calculated 5-year moving averages of PM$_{2.5}$ for each individual, treating these as time-varying exposure windows. For example, a participant entering Medicare in 2000 and starting follow-up in 2005 following the 5-year clean period without AD, would have PM$_{2.5}$ estimates based on average concentrations from 2000 to 2004, for their first follow-up year (2005). We did not have ZIP-code–level PM$_{2.5}$ exposure data prior to 2000 and were therefore unable to estimate earlier-life exposures for Medicare participants. For the mediation analyses, exposure was estimated in a slightly different way to ensure the temporal sequence required for mediation (see Statistical analysis below for details).

## Outcome assessment

In this study, we focused on AD as the outcome. We defined AD based on the date of the first recorded diagnosis in the Medicare data, using either AD diagnosis codes or dementia diagnosis followed by AD, under the assumption that AD may be under-recorded initially or diagnosed late [20,25]. Among participants with dementia preceding AD, the median duration between the first dementia diagnosis and the subsequent AD diagnosis was approximately 2 years (interquartile range [IQR]: 1–4 years). The CCW algorithm identifies dementia and AD diagnoses by analyzing Medicare claims, including home healthcare, skilled nursing facility claims, carrier claims (primarily for doctor visits), and inpatient and outpatient hospital records. This algorithm has been validated and shown reasonable accuracy in classifying diseases in previous studies [28,29], utilizing International Classification of Diseases (ICD) codes: ICD-9 code 331.0 and ICD-10 codes G30.0, G30.1, G30.8, and G30.9.

## Comorbidities assessment

We examined three comorbidities—hypertension, stroke, and depression—as potential mediators or modifiers in the relationship between $PM_{2.5}$ exposure and AD, based on previous findings that link these conditions strongly to higher AD rates [20] and its recognition as a potentially modifiable risk factor for dementia [30]. Comorbidities were defined as the first recorded diagnosis of hypertension, stroke, or depression, as identified by the Medicare CCW. The specific ICD codes and algorithm for diagnosis of these three comorbidities in the Medicare CCW are presented in S1 Table.

## Covariates

We adjusted for several covariates, including study year and individual-level characteristics (age at entry, year of entry, sex, race, and Medicaid eligibility), as well as a number of ZIP code-level average SES indicators. Although Medicaid dual eligibility does not directly measure education or employment, it serves as a validated proxy for low income and financial hardship and is widely used as an indicator of individual-level socioeconomic disadvantage in Medicare-based studies [31]. Ecological or area-level variables included healthcare capacity indicators (number of hospitals) in the zip code, and a geographical region indicator representing five U.S. regions. Additionally, we considered county-level behavioral risk factors (smoking rates [% of population ever smokers] and mean body mass index [BMI]) and ZIP code-level SES variables, such as population density, median household income, the percentage of Black individuals, the percentage of individuals with less than a high school education, percentage of the population below the poverty line, and percentage of the population living in renting house. These covariates were selected based on prior literature indicating that they could be associated with $PM_{2.5}$ exposure, comorbidities, and AD, and therefore may confound the observed relationships between $PM_{2.5}$ and AD, as well as between $PM_{2.5}$ and comorbidities [20,25,32]. Individual-level data were sourced from the Medicare denominator file, healthcare capacity information from the American Hospital Association Annual Survey Database [33], county-level behavioral risk factors from the Behavioral Risk Factor Surveillance System [34], and ZIP code-level SES data from the U.S. Census [35,36] and the American Community Survey [37].

## Statistical analysis

There was no prospective protocol, and the analysis plan was as follows. Stratified Cox proportional hazards models with generalized estimating equations (GEE) were utilized to examine the associations between time-varying $PM_{2.5}$ exposure (5-year moving averages) and AD. The GEE model accounted for residual autocorrelation within ZIP codes using robust standard errors. Analyses were stratified by 1-year categories of age at study entry, race, sex, and Medicaid eligibility, and adjusted for the aforementioned covariates, including calendar year and time-updated socioeconomic indicators, to account for temporal changes in population composition and minimize potential bias related to differences in the Medicare/Medicaid population by SES over time. We explored how comorbidities influence the relationship between $PM_{2.5}$ exposure and AD in two ways. First, we tested the hypothesis of effect modification by comorbidities. Specifically, we

posited that the effect of $PM_{2.5}$ (5-year moving averages) on AD varies based on the presence of comorbidities. To test this, we stratified analyses by comorbidities occurring at any time before the AD diagnosis. The Wald test was used to estimate $P$-values for interaction. Second, we also hypothesized that comorbidities may mediate the association between $PM_{2.5}$ and AD.

To examine the potential mediating role of comorbidities, we created three distinct cohorts, ensuring that conditions such as hypertension, stroke, and depression developed after $PM_{2.5}$ exposure and before the onset of AD. Participants could contribute to more than one cohort if they developed multiple comorbidities, but each comorbidity was modeled independently as a potential mediator. Joint mediation across multiple comorbidities was not evaluated.

To then assess the possibility of mediation, we first assessed the association between time-varying $PM_{2.5}$ exposure (5-year moving averages) and comorbidities during the follow-up using Cox proportional hazards models. Next, we tested whether individuals with comorbidities were more likely to develop incident AD, also using Cox proportional hazards models.

We assumed a temporal causal pathway of $PM_{2.5} \rightarrow$ comorbidity$\rightarrow$AD. To minimize the potential for reverse causation (AD$\rightarrow$comorbidity), comorbidities were required to occur sometime prior to AD diagnosis. Comorbidities, if present, were required to develop between the first year of entry and the occurrence of AD, to ensure they were potential mediators. This temporal structure was designed to ensure proper temporal ordering rather than to imply short-term causal effects. In practice, these comorbidities were first diagnosed in our mediation analysis several years before the first recorded AD diagnosis in the Medicare data. To ensure that the comorbidities followed exposure, exposure for comorbidity analysis was defined as fixed, and based on the $PM_{2.5}$ level of the participant's zip code at the year of the start of the clean period. Furthermore, participants were required not to have moved out of their original zip code during follow-up (the percentage of non-movers in our cohort was 84.6%), following the method of a previous analysis of mediation [23]. By requiring no movement during follow-up, we increased the likelihood that exposure at entry year was representative of later exposure. Using exposure at the start of the clean period and restricting the cohort to individuals without comorbidities at year of entry helped ensure that comorbidities developed after exposure.

Participants were excluded from the mediation analysis if: 1) they had a history of comorbidities at the start of the clean period (i.e., at the same time that exposure was defined to ensure that the comorbidity followed exposure) or 2) they were diagnosed with the comorbidity in the same year as their AD diagnosis (as we could not know if the comorbidity preceded the AD, a requirement for possible mediation). These conditions fulfilled the mediation criteria that the exposure is associated with subsequent comorbidities and that comorbidities are associated with subsequent AD.

After assessing the possibility of mediation and finding it to be plausible, we then analyzed the mediation effect using the product method, and applied two-way decomposition causal mediation methods, considering potential interactions between exposure and mediator [38]. We fitted two Cox proportional hazards models to determine the direct effect (DE), indirect effect (IE), total effect (TE), and the proportion mediated. Our mediation analysis assumes no unmeasured confounding for the exposure–outcome, exposure–mediator, or mediator–outcome pathways, and no mediator–outcome confounder that is itself influenced by the exposure. Here, "a" represents $PM_{2.5}$, "m" represents the mediator (incident comorbidity event, hypertension, stroke, or depression), and "c" represents the set of covariates described above.

Equation (1) (the outcome model):

$$\lambda(AD|a, m, c) = \lambda_0 \exp(\theta_1 a + \theta_2 m + \theta_3 am + \theta_4' c) \tag{1}$$

Equation (2) (the mediator model):

$$\lambda(M|PM_{2.5}, c) = \lambda_0 \exp(\beta_1 a + \beta_2' c) \tag{2}$$

When both the mediator and outcome are binary, the DE, IE, TE, and proportion mediated are calculated using the formulas derived from VanderWeele (2016) [38], which account for the interactions between PM$_{2.5}$ exposure and comorbidities (see below). In VanderWeele's approach, these formulas were based on two logistic regression models. However, in our study, we use two Cox proportional hazards models. As a result, compared to the formulas in VanderWeele (2016) [38], we omit the $\beta_0$ term, effectively setting β0 to 0 in our models. In the equation, "a" represents the exposure level of interest, "a*" represents the reference exposure level, and "c" denotes the mean of the covariates used for standardization. Accordingly, we set a = 1, a* = 0, and c equal to the covariate means to estimate the controlled direct and indirect effects under this framework.

$$DE = \frac{\exp(\theta 1a)\left\{1 + \exp\left(\theta 2 + \theta 3a + \beta 0 + \beta 1a^* + \beta 2'c\right)\right\}}{\exp(\theta 1a^*)\left\{1 + \exp\left(\theta 2 + \theta 3a^* + \beta 0 + \beta 1a^* + \beta 2'c\right)\right\}}$$

$$IE = \frac{\left\{1 + \exp\left(\beta 0 + \beta 1a^* + \beta 2'c\right)\right\}\{1 + \exp(\theta 2 + \theta 3a + \beta 0 + \beta 1a + \beta 2'c)\}}{\{1 + \exp(\beta 0 + \beta 1a + \beta 2'c)\}\{1 + \exp\left(\theta 2 + \theta 3a + \beta 0 + \beta 1a^* + \beta 2'c\right)\}}$$

$$TE = DE * IE$$

$$\text{Proportion mediated} = \frac{DE * (IE - 1)}{DE * IE - 1}$$

Our results are presented for a one-unit increase in PM$_{2.5}$. We calculated 95% confidence intervals (CIs) for TE, DE, IE, and mediated proportions using bootstrapping with 100 resamples. We additionally performed a sensitivity analysis restricting the outcome definition to participants with a direct AD diagnosis only, excluding cases where dementia diagnosis preceded AD. This analysis was conducted to evaluate the robustness of our primary findings to alternative case definitions and potential diagnostic delays. All statistical analyses were performed using R software (version 4.2.3) on the high-performance computing (HPC) cluster at Emory University, with a two-sided significance level set at $P < 0.05$. This study is reported as per the Strengthening the Reporting of Observational Studies in Epidemiology (STROBE) guideline (S1 STROBE Checklist).

## Results

Table 1 summarizes the demographic characteristics of the overall population and three cohorts after a 5-year clean period. The study included 27,763,593 participants, with 2,997,902 (10.8%) developing AD. The mean age at entry after a 5-year clean period was approximately 76 years, with a median follow-up of 6 years. Over 57.9% of participants were female, the majority were White (over 89.2%), and more than 84.1% were ineligible for Medicaid. The 5-year average PM$_{2.5}$ concentration was 10.1 μg/m³ (IQR: 3.8 μg/m³). The prevalence of hypertension, stroke, and depression after a 5-year clean period were 86.9%, 23.2%, and 34.9%. After excluding participants with a history of comorbidities at the start of the clean period and those diagnosed with both comorbidities and AD in the same year, the final cohorts consisted of 13,827,457 participants for hypertension, 26,517,760 for stroke, and 25,222,439 for depression (study flowchart in Fig 1). The hypertension cohort comprised 13.8 million individuals, accumulating 93.3 million person-years of follow-up, during which approximately 1.3 million participants developed AD events (9.3%) and 10.2 million were diagnosed with hypertension during the follow-up (73.8%). Due to the high prevalence of hypertension in the Medicare population (86.9%) and our requirement that any hypertension cases in the cohort

**Table 1. Descriptive statistics for the study population and distribution of air pollution from 2000 to 2018 with 5-year clean period [n (%) or mean (SD)].**

| Variables | Overall population | PM$_{2.5}$→hypertension→AD cohort | PM$_{2.5}$→stroke→AD cohort | PM$_{2.5}$→depression→AD cohort |
|---|---|---|---|---|
| **Characteristics** | | | | |
| Number of the total population | 27,763,593 (100) | 13,827,457 (100) | 26,517,760 (100) | 25,222,439 (100) |
| Total person-years | 178,701,891 (100) | 93,323,675 (100) | 172,088,904 (100) | 164,885,974 (100) |
| Number of AD | 2,997,902 (10.8) | 1,285,845 (9.3) | 2,609,276 (9.8) | 2,341,253 (9.3) |
| Number of comorbidities | | | | |
| Hypertension | 24,126,083 (86.9) | 10,200,634 (73.8) | | |
| Stroke | 6,442,649 (23.2) | | 5,212,504 (19.7) | |
| Depression | 9,685,767 (34.9) | | | 7,177,542 (28.5) |
| Median follow-up years | 6 | 6 | 6 | 6 |
| Age at entry (years) | 75.72 (6.27) | 75.02 (5.84) | 75.56 (6.19) | 75.67 (6.25) |
| 65–74 | 16,046,375 (57.8) | 8,660,282 (62.6) | 15,626,051 (58.9) | 14,660,493 (58.1) |
| 75–114 | 11,717,218 (42.2) | 5,167,175 (37.4) | 10,891,709 (41.1) | 10,561,946 (41.9) |
| Sex | | | | |
| Male | 11,676,349 (42.1) | 6,224,683 (45.0) | 11,165,492 (42.1) | 10,992,719 (43.6) |
| Female | 16,087,244 (57.9) | 7,602,774 (55.0) | 15,352,268 (57.9) | 14,229,720 (56.4) |
| Race | | | | |
| White | 24,768,655 (89.2) | 12,603,646 (91.1) | 23,671,951 (89.3) | 22,437,237 (89.0) |
| Black | 1,861,196 (6.7) | 662,762 (4.8) | 1,757,129 (6.6) | 1,741,098 (6.9) |
| Other[a] | 1,133,742 (4.1) | 561,049 (4.1) | 1,088,680 (4.1) | 1,044,104 (4.1) |
| Medicaid eligibility | | | | |
| Ineligible | 23,336,913 (84.1) | 12,012,474 (86.9) | 22,406,320 (84.5) | 21,406,135 (84.9) |
| Ever eligible | 4,426,680 (15.9) | 1,814,983 (13.1) | 4,111,440 (15.5) | 3,816,304 (15.1) |
| Regions | | | | |
| Midwest | 7,236,505 (26.1) | 3,661,476 (26.5) | 6,924,608 (26.1) | 6,574,527 (26.1) |
| Northeast | 4,992,988 (18.0) | 2,324,232 (16.8) | 4,757,589 (17.9) | 4,529,179 (18.0) |
| Southeast | 8,840,334 (31.8) | 4,196,241 (30.3) | 8,418,875 (31.7) | 8,022,270 (31.8) |
| Southwest | 2,895,558 (10.4) | 1,520,182 (11.0) | 2,767,475 (10.4) | 2,630,526 (10.4) |
| West | 3,798,208 (13.7) | 2,125,326 (15.4) | 3,649,213 (13.8) | 3,465,937 (13.7) |
| **Exposure (µg/m³)[b]** | | | | |
| PM$_{2.5}$, mean (IQR) | 10.1 (3.8) | 9.9 (3.8) | 10.1 (3.8) | 10.1 (3.8) |

Abbreviations: AD, Alzheimer's disease; IQR: interquartile range; PM$_{2.5}$, fine particulate matter; SD, standard deviation.

[a]Other included Asian, Hispanic, American Indian, or Alaskan Native, and unknown.

[b]Exposure was estimated as mean exposure in the prior 5-year window.

occur during the follow-up, many potential participants were excluded. In the stroke cohort, which included 26.5 million individuals and recorded 172.1 million person-years of follow-up, around 2.6 million individuals experienced AD events (9.8%) and 5.2 million were diagnosed with stroke (19.7%). The depression cohort, consisting of 25.2 million individuals and 164.9 million person-years of follow-up, reported approximately 2.3 million AD events (9.3%) and 7.2 million depression events (28.5%) during the follow-up period.

Table 2 presents the association between exposure to PM$_{2.5}$ and AD. In the overall population, a higher hazard ratio (HR) for AD was observed with each IQR increase in the 5-year average concentrations of PM$_{2.5}$, with an HR of 1.085 (95% CI: 1.078, 1.091), after adjusting for all covariates and three comorbidities (full model results can be found in S2 Table). In the stratified analyses by comorbidities to assess effect modification, individuals with stroke experienced a

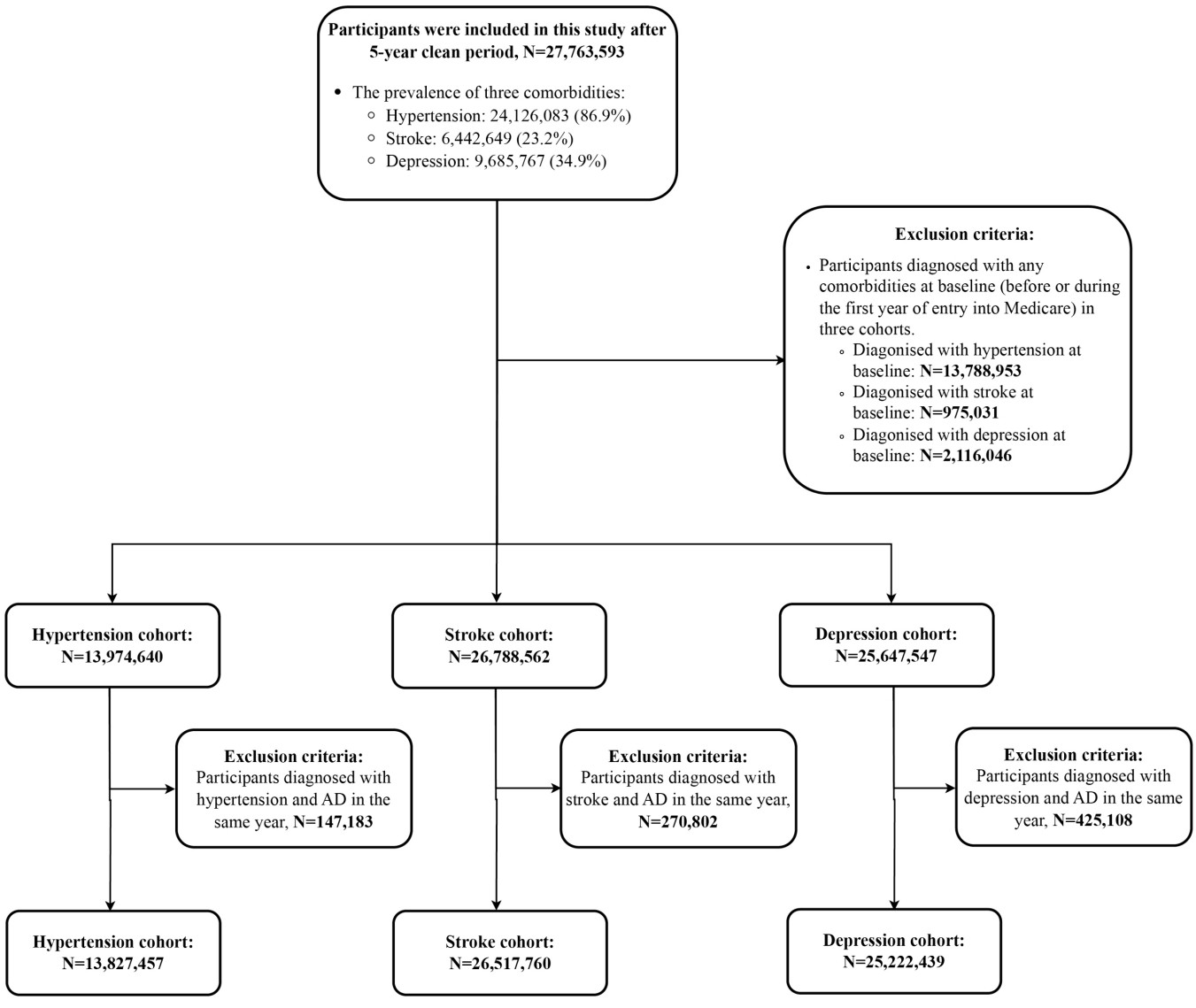

**Fig 1. Flowchart of study population selection and exclusions.** Abbreviations: AD, Alzheimer's disease.

somewhat higher risk of AD linked to per IQR increase in PM$_{2.5}$ exposure, with HR of 1.105 (95% CI: 1.096, 1.114) compared to HRs of 1.088 (95% CI: 1.082, 1.095) for those without stroke (*P* for interaction = 0.004). No significant statistical differences were observed in the effect of PM$_{2.5}$ on AD between individuals with and without hypertension or depression. These associations remained consistent after excluding participants with a history of comorbidities at baseline or those diagnosed with both comorbidities and AD in the same year (S3 Table).

Fig 2 shows that the conditions for mediation were present, as per IQR increase in PM$_{2.5}$ was associated with the three comorbidities, and in turn, the comorbidities were associated with AD. Table 3 presents the results of the causal mediation analysis (using a one-unit change in PM$_{2.5}$ as the metric, not the IQR). Nearly all the overall associations were dominated by direct associations, with minimal evidence that the effect of PM$_{2.5}$ on AD was mediated through comorbidities. Specifically, approximately 1.6% of the effect was mediated by hypertension, about 4.2% by stroke, and around 2.1%

**Table 2. Subgroup analysis by comorbidities of hazard ratios and 95% CIs of per IQR increase in PM$_{2.5}$ associated with AD.**

|  | HR (95% CI) | *P*-value for interaction[b] |
|---|---|---|
| **Overall population**[a] | 1.085 (1.078, 1.091) | – |
| Without hypertension | 1.099 (1.089, 1.109) | 0.706 |
| With hypertension | 1.097 (1.090, 1.104) | |
| Without stroke | 1.088 (1.082, 1.095) | 0.004 |
| With stroke | 1.105 (1.096, 1.114) | |
| Without depression | 1.092 (1.085, 1.098) | 0.905 |
| With depression | 1.092 (1.084, 1.101) | |

Abbreviations: AD, Alzheimer's disease; CI, confidence interval; HR, hazard ratios; IQR: interquartile range; PM$_{2.5}$, fine particulate matter.

[a]Model was conducted for the overall population.

[b]*P*-value for interaction term was estimated by the Wald test.

Exposure was estimated as mean exposure in the prior 5-year window.

All those three comorbidities occur before or in the same year as the diagnosis of AD (e.g., the "no hypertension" group composed of those who never had hypertension prior to their first diagnosis of AD).

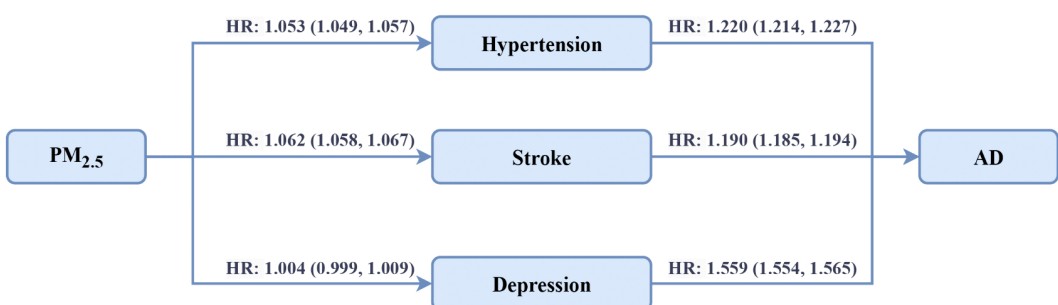

**Fig 2. Illustration of the association of PM$_{2.5}$ exposure, comorbidities, and incident AD.** Hazard ratios and 95% confidence intervals were calculated using Cox proportional hazards regression models to analyze the association between per interquartile range increase in PM$_{2.5}$ exposure and comorbidities (left side), as well as the association between comorbidities and AD (right side). Abbreviations: AD, Alzheimer's disease; CI, confidence intervals; HR, hazard ratios; IQR: interquartile range; PM$_{2.5}$, fine particulate matter.

by depression. In a sensitivity analysis restricted to participants with a direct AD diagnosis only (excluding dementia-preceded cases), both stratification and mediation results were consistent with the main findings (S4, S5 Tables).

## Discussion

In this nationwide cohort study in the U.S., we evaluate the dual role of multiple comorbidities as both effect modifiers and potential mediators in the relationship between PM$_{2.5}$ exposure and incident AD. We identified positive associations of exposure to PM$_{2.5}$ with the incidence of AD, as well as positive associations of PM$_{2.5}$ with hypertension, depression, and stroke. All three comorbidities were linked to an increased risk of AD, suggesting the possibility of mediation by these comorbidities. However, we found a minimal IE of PM$_{2.5}$ on incident AD through hypertension, stroke, and depression, indicating that PM$_{2.5}$ increases the risk of developing AD largely independent of these comorbidities. The mediation effect was slightly stronger among individuals with stroke, suggesting increased vulnerability in this subgroup.

**Table 3. Two-way decomposition of the association of per 1 µg/m³ increase in PM$_{2.5}$ with incident AD by comorbidities using causal mediation analysis.**

|  | Hypertension | Stroke | Depression |
|---|---|---|---|
| Direct effect | 1.015 (1.014, 1.015) | 1.015 (1.015, 1.016) | 1.015 (1.015, 1.016) |
| Indirect effect | 1.0002 (1.0002, 1.0003) | 1.0007 (1.0006, 1.0007) | 1.0003 (1.0003, 1.0004) |
| Total effect | 1.015 (1.015, 1.015) | 1.016 (1.016, 1.017) | 1.016 (1.015, 1.016) |
| Proportion Mediated, % | 1.6 | 4.2 | 2.1 |

Abbreviations: AD, Alzheimer's disease; PM$_{2.5}$, fine particulate matter.

To ensure that the comorbidities followed exposure contributed to the development of AD in our mediation analysis, exposure was defined as fixed and based on the participant's zip code at the time of entry into Medicare, and participants were required not to have moved out of their original zip code during follow-up.

Our findings are consistent with a substantial body of research indicating that exposure to PM$_{2.5}$ is associated with an increased risk of AD, with comparable effect sizes across studies. For instance, in a large national cohort study using the same Medicare database as our research, Shi and colleagues [25] reported an HR of 1.078 for AD per IQR (3.2 µg/m³) increase in 5-year average PM$_{2.5}$. A systematic review and meta-analysis [39] found a pooled HR of 3.26 (95% CI: 1.20–5.31) for AD per 10 µg/m³ increase in PM$_{2.5}$, a result echoed by another meta-analysis [40] that reported a similar HR of 3.26 (95% CI: 0.84–12.74).

Additionally, our results support existing evidence of the association between PM$_{2.5}$ exposure and the risks of depression [18], hypertension [41], and stroke [42]. All these three comorbidities are also significantly linked to a higher risk of AD in our data and previous research using the same Medicare data [20]. As noted, these relationships suggest the possibility of mediation. Our work here adds to the existing mediation literature by formally modeling and quantifying the relationships between PM$_{2.5}$, these comorbidities, and AD in a large national cohort, and requiring a clear temporal ordering of exposure, mediators, and outcomes.

However, our analysis revealed that the proportion of the association between PM$_{2.5}$ and incident AD mediated by these comorbidities was quite small, with mediated proportions of 1.6% for hypertension, 2.1% for depression, and 4.2% for stroke. While prior research has highlighted that neurodegenerative processes may begin much earlier in life, our study complements this evidence by quantifying the contribution of late-life exposures and comorbidities to clinically diagnosed AD in older adults. Our finding aligns with a nationally representative cohort study involving 27,857 U.S. participants, which found no evidence that hypertension or stroke served as mediators in the relationship between PM$_{2.5}$ and incident dementia [23]. Similarly, a recent study of 2,564 older adults in the U.S. reported no mediation through hypertension for the effect of PM$_{2.5}$ on either AD or vascular dementia [24]. In contrast, in a Canadian cohort study of 34,391 older residents, Ilango and colleagues found that 21% of the total association between PM$_{2.5}$ and dementia could be attributed to CVD [21]. One key difference between their findings and ours likely lies in their failure to account for exposure-mediator interactions in their initial results. When they included these interactions in supplementary analyses, the estimated mediated proportion decreased significantly from 21% to 5%, which is more in line with our results. In a Swedish cohort of 2,927 older participants, Grande and colleagues reported that 49% of the association between PM$_{2.5}$ and dementia could be attributed to an indirect pathway through baseline stroke, using generalized structural equation modeling [22]. Their relatively small sample size may lead to imprecise mediation effects. Additionally, they calculated the IE based on the associations among baseline exposure, baseline mediators, and outcomes using two logistic regressions, which may not effectively capture the timing of events, potentially introducing bias.

We found that the increased risk of AD associated with PM$_{2.5}$ exposure was slightly more pronounced among individuals with stroke, suggesting a potential role of vascular vulnerability. In support of our findings, a Swedish National study among 2,253 participants indicated that the detrimental relationship between air pollution and cognitive decline was

worsened by the presence and development of cerebrovascular diseases, including stroke [43]. A cohort study of 2,927 Swedish participants also reported that CVD, particularly heart failure and ischemic heart disease, appeared to amplify the negative association of air pollution with dementia [22]. The observed effect modification by stroke may reflect an underlying biological vulnerability in cerebrovascular pathways. Stroke-related neurovascular damage can compromise the blood–brain barrier, facilitating the translocation of $PM_{2.5}$ particles or their associated inflammatory mediators into the brain. In turn, this can exacerbate neuroinflammation, oxidative stress, and amyloid-beta accumulation—hallmarks of Alzheimer's pathology [44,45]. Additionally, individuals with stroke may experience altered cerebral perfusion or impaired glymphatic clearance, both of which may interact synergistically with environmental insults to accelerate neurodegeneration [46]. These findings align with emerging mechanistic studies that highlight shared pathways between cerebrovascular injury and pollution-induced neural damage [44,45,47]. The stronger $PM_{2.5}$–AD association observed among individuals with stroke supports a synergistic model in which cerebrovascular pathology increases vulnerability to pollution-related neurodegenerative processes, rather than indicating that vascular disease alone drives the observed association.

This study has several key strengths. First, it is the first large-scale national cohort study to simultaneously evaluate the roles of hypertension, stroke, and depression in the association between air pollution and incident AD among US Medicare enrollees. The substantial sample size provides strong statistical power to accurately assess these effects. Second, we focused on post-exposure incident cases of hypertension, stroke, and depression as potential mediators, rather than their baseline prevalence, to capture the temporal sequence of exposure, mediators, and AD. This approach allowed us to better understand how air pollution may influence the development of these intermediates over time and, in turn, lead to incident AD, while accounting for the chronological order of these events. Third, our use of mediation analysis incorporated exposure–mediator interactions, mitigating potential bias in effect estimates, particularly when interactions are significant.

Despite these strengths, our study has several limitations. First, while the exposure prediction model performs well, there is a potential measurement error because $PM_{2.5}$ levels were assigned based on ZIP code instead of individual residential addresses. However, other recent work has suggested that measurement error in our modeled exposure data would not have a strong effect [48,49]. Additionally, our exposure assessment captured only outdoor ambient $PM_{2.5}$ concentrations and did not account for household or occupational exposures (e.g., cooking, heating, or workplace sources), which may contribute to total exposure but are unavailable at the national scale. These unmeasured sources could lead to exposure misclassification, although existing data indicate that ambient and total personal exposures are reasonably well correlated ($r = 0.60$) based on U.S. data [50]. Such misclassification is likely non-differential with respect to AD outcomes and would bias our estimates toward the null. Furthermore, we examined 5-year average exposure immediately preceding disease onset and were unable to estimate exposures earlier in life due to the lack of historical exposure data. It is likely that the disease process began earlier, and our findings may therefore reflect the correlation of relatively recent exposure with past exposure levels. Second, using administrative records to identify disease may lead to outcome misclassification. Nevertheless, Medicare inpatient and outpatient claims provide near-complete coverage of medical encounters for fee-for-service beneficiaries aged ≥65 years from both public and private healthcare providers. Previous literature has shown that AD diagnoses in Medicare data have high specificity (95%) and moderate sensitivity (64%), supporting their reliability for population-based research [29]. Moreover, in our previous work on air pollution and AD, Shi and colleagues [25] conducted two sensitivity analyses to address this issue: one using a linear regression model based on rates, and the other using prior estimates of Medicare's sensitivity and specificity to adjust the case counts. They found that correcting for misclassification of outcome did not significantly alter the results, suggesting that such misclassification likely biased our findings slightly toward the null. Third, although we accounted for multiple covariates, some data were only available at the ZIP code level. Consequently, we were able to adjust for some risk factors, such as smoking and BMI, only at the area level, which may have introduced some residual confounding in the $PM_{2.5}$–AD or $PM_{2.5}$–comorbidity associations, particularly if these unmeasured individual behaviors were correlated with air pollution levels. However, we adjusted for

individual-level Medicaid eligibility and race, both important indicators of SES, as well as other key individual-level confounders, i.e., age and sex. Moreover, validation literature has shown that area-based socioeconomic indicators yield results consistent in direction and slightly attenuated compared with those using individual-level SES data, suggesting that any residual misclassification would likely bias our results only slightly toward the null [51–53]. Overall, we believe that our control for a large number of individual- and area-level potential confounders was reasonably thorough,

and that confounding was unlikely to markedly distort our findings for either the PM$_{2.5}$/AD associations or the PM$_{2.5}$/mediator associations.

In conclusion, PM$_{2.5}$ exposure was associated with increased AD risk in a large national cohort, primarily through direct pathways rather than mediation by common comorbidities. Notably, the association was modestly stronger among individuals with stroke, suggesting heightened vulnerability in this subgroup. Our findings suggest that reducing air pollution could benefit cognitive health broadly across older adults, while targeted interventions may be especially important for those with cerebrovascular disease or multiple chronic conditions.

## Supporting information

**S1 STROBE Checklist. STROBE Statement—Checklist of items that should be included in reports of cohort studies.** This checklist is reproduced from the STROBE Statement (Strengthening the Reporting of Observational Studies in Epidemiology) and is licensed under the Creative Commons Attribution 4.0 International (CC BY 4.0). Source: https://www.strobe-statement.org/.
(DOCX)

**S1 Table. ICD-codes for comorbidities used in CCW database.**
(DOCX)

**S2 Table. Association of PM$_{2.5}$ and confounders with incident AD in the complete model.**
(DOCX)

**S3 Table. Subgroup analysis by comorbidities of hazard ratios and 95% CIs of per IQR increase in PM$_{2.5}$ associated with AD in three different cohorts.**
(DOCX)

**S4 Table. Subgroup analysis by comorbidities of hazard ratios and 95% CIs of per IQR increase in PM$_{2.5}$ associated with AD, using a restricted outcome definition based on direct AD diagnoses only.**
(DOCX)

**S5 Table. Two-way decomposition of the association of per 1 μg/m$^3$ increase in PM$_{2.5}$ with incident AD by comorbidities using causal mediation analysis under a restricted outcome definition based on direct AD diagnoses only.**
(DOCX)

## Acknowledgments

We would like to especially thank the Centers for Medicare & Medicaid Services for providing access to the Medicare claims data used in this study.

The content is solely the responsibility of the authors and does not necessarily represent the official views of the National Institutes of Health. It is subject to the NIH Public Access Policy. Through acceptance of this federal funding, NIH has been given a right to make this manuscript publicly available in PubMed Central upon the Official Date of Publication, as defined by NIH.

## Author contributions

**Conceptualization:** Yanling Deng, Yang Liu, Kyle Steenland.

**Data curation:** Yanling Deng, Yang Liu, Ke Xu, Kyle Steenland.

**Formal analysis:** Yanling Deng.

**Funding acquisition:** Yang Liu, Kyle Steenland.

**Investigation:** Yanling Deng.

**Methodology:** Yanling Deng, Hua Hao.

**Project administration:** Yang Liu, Kyle Steenland.

**Resources:** Yang Liu, Kyle Steenland.

**Software:** Yanling Deng, Ke Xu, Qiao Zhu.

**Supervision:** Yang Liu, Kyle Steenland.

**Validation:** Hua Hao, Kyle Steenland.

**Visualization:** Yanling Deng.

**Writing – original draft:** Yanling Deng.

**Writing – review & editing:** Yanling Deng, Yang Liu, Hua Hao, Ke Xu, Qiao Zhu, Haomin Li, Tszshan Ma, Kyle Steenland.

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
