## [Editor Report · Decision Letter 0]

4 Jul 2025

Dear Dr Deng,

Thank you for submitting your manuscript entitled "The Role of Comorbidities in the Associations between Air Pollution and Alzheimer's Disease: A National Cohort Study in the American Medicare Population" for consideration by PLOS Medicine.

Your manuscript has now been evaluated by the PLOS Medicine editorial staff as well as by an academic editor with relevant expertise and I am writing to let you know that we would like to send your submission out for external peer review.

For clinical studies, please upload a copy of your trial study protocol as a supporting information file. The study protocol should be the version submitted for approval to the institutional review board or ethics committee, should include any amendments to the study protocol, as well as the date of their approval by the institutional review or ethics committee. Please also detail any deviations from the study protocol in the Methods section of your manuscript. The editors will consider the protocol and study conduct prior to a final decision for external review.

Please re-submit your manuscript within two working days, i.e. by Jul 08 2025 11:59PM.

Kind regards,

Suzanne De Bruijn, PhD

Senior Editor

PLOS Medicine

---

## [Decision Letter · Decision Letter 1]

9 Oct 2025

Dear Dr Deng,

Many thanks for submitting your manuscript "The Role of Comorbidities in the Associations between Air Pollution and Alzheimer's Disease: A National Cohort Study in the American Medicare Population" (PMEDICINE-D-25-02362R1) to PLOS Medicine. The paper has been reviewed by subject experts and a statistician; their comments are included below and can also be accessed here: [LINK]

As you will see, the reviewers showed interest in your study, but also raised concerns about potential confounders. Furthermore, there were several requests for clarifications and more detail. After discussing the paper with the editorial team and an academic editor with relevant expertise, I'm pleased to invite you to revise the paper in response to the reviewers' comments. We plan to send the revised paper to some or all of the original reviewers, and we cannot provide any guarantees at this stage regarding publication.

We ask that you submit your revision by Oct 30 2025 11:59PM. However, if this deadline is not feasible, please contact me by email, and we can discuss a suitable alternative.

Don't hesitate to contact me directly with any questions (sbruijn@plos.org).

Best regards,

Suzanne

Suzanne De Bruijn, PhD

Associate Editor

PLOS Medicine

sbruijn@plos.org

Comments from the reviewers:

Reviewer #1: "The Role of Comorbidities in the Associations between Air Pollution and Alzheimer's Disease: A National Cohort Study in the American Medicare Population" investigates the effect of known Alzheimer's Disease (AD) comorbidities, in air pollution as quantified by PM2.5. Stratified and mediation analyses were performed on a nationwide cohort of some 28 million elderly subjects (with 3 million incident AD cases) from 2000 to 2018, to estimate the mediation effect of comorbidities, on the association between PM2.5 and AD. In general, minimal mediation effects were found, suggesting that the association between PM2.5 exposure and AD is largely through direct pathways.

Some issues might be considered:

1. In the Introduction, the lack of studies investigating the role of comorbidities as intermediaries between PM2.5 and AD is claimed. However, in general, it appears that the following analysis might not be comprehensive enough to draw meaningful conclusions, as acknowledged in the limitations on the presence of large numbers of potential individual confounders (which would seem to possibly have significantly larger impacts on AD prevalence, than the relatively low impact of PM2.5). The value of the study might thus be further justified.

2. In Section 2.1, it is stated that the open cohorts used had subjects able to enter and leave at any time from 2000 to 2018. It might be clarified as to whether this might bias the cohorts (e.g. correlation between increased socioeconomic status and decreased Medicare/Medicaid participation).

3. The completeness of the data in the cohort might be clarified. In particular, would a diagnosis of AD necessarily be included in the two databases used, or might such data be missing if the patient sought treatment at private institutions?

4. In Section 2.3, it is stated that AD is defined based on either AD diagnosis codes, or dementia diagnosis followed by AD. It might be clarified as to the distribution of duration between dementia diagnosis and AD, and whether the strict application of AD diagnosis as outcome data would materially affect the findings.

5. In Section 2.6, the statistical analysis using stratified Cox proportional hazard models with generalized estimating equations is described. It might be clarified as to whether any assumptions regarding (direction of) causality between the comorbidities and outcome (in particular AD -> comorbidity) were made, regarding the models. It is stated that two-way decomposition causal mediation methods were applied, with the results presented in Table 3. However, this does not appear to show the direction(s) of causality. This might be clarified.

6. In Section 2.6, it is mentioned that three distinct cohorts were created, one for each comorbidity. It might be clarified as to whether subjects may possess multiple comorbidities, and if so, whether they would be included in multiple cohorts. In that latter case, how is mediation on multiple comorbidities performed?

7. In Section 2.6, the assumptions on variables a, a* and c might be explained further.

Reviewer #2: Please see in the attachment

Reviewer #3: The authors report a very interesting large study using administrative data from Medicare beneficiaries in the United States to investigate the association between exposure to air pollution and incident Alzheimer's disease. While other studies have done this, the authors evaluated whether the association was mediated by hypertension, stroke and depression. The latter three risk factors for dementia area also impacted by exposure to air pollution. This study therefore attempts to shed light on the direct vs indirect effect of air pollution on dementia which is important for elucidating potential mechanisms and also interventions. The paper has several strengths. The sample size is huge and the study period is 18 years. The research question is important because of the large number of people exposed to air pollution and the high prevalence of the three risk factors.

There are some limitations of the study which reduce the clarify of findings their potential significance.

1. The authors mention adjusting for covariates including measures of socioeconomic status such as percentage of population below the poverty line, percentage with less than high school education as these covariates have been associated with AD and PM. It appears that these covariates were not adjusted for individuals, but for locations. This is a limitation in the methodology - presumably due to lack of demographic data in the Medicare data. Sociodemographic variables are the biggest confounders of the association between the key variables ie people with low education and low SES are more likely to develop chronic diseases, and AD so the way these are handled in the analysis is very important for interpretation of results. Further information needs to be provided on the methods used to adjust for these variables and the results for the associations between these confounders and the exposure and outcome should be reported. This will assist in interpreting the findings.

2. Was there any demographic information in the Medicare database eg. on education etc. that could have provided individual level SES variables?

3. Is there way of evaluating aggregation bias caused by using the population level measures of SES? What degree of misclassification can occur using this methodology? I think this needs to be explained far better to assure readers that the methodology for this adjustment is adequate and the associations reported are reliable.

---

* Please upload any figures associated with your paper as individual TIF or EPS files with 300dpi resolution at resubmission; please read our figure guidelines for more information on our requirements: http://journals.plos.org/plosmedicine/s/figures. While revising your submission, we strongly recommend that you use PLOS's NAAS tool (https://ngplosjournals.pagemajik.ai/artanalysis) to test your figure files. NAAS can convert your figure files to the TIFF file type and meet basic requirements (such as print size, resolution), or provide you with a report on issues that do not meet our requirements and that NAAS cannot fix.

After uploading your figures to PLOS's NAAS tool - https://ngplosjournals.pagemajik.ai/artanalysis, NAAS will process the files provided and display the results in the "Uploaded Files" section of the page as the processing is complete.

If the uploaded figures meet our requirements (or NAAS is able to fix the files to meet our requirements), the figure will be marked as "fixed" above. If NAAS is unable to fix the files, a red "failed" label will appear above.

When NAAS has confirmed that the figure files meet our requirements, please download the file via the download option, and include these NAAS processed figure files when submitting your revised manuscript.

* In you data availability statement, please include how the data can be accessed, including contact details (email or URL).

* In you financial disclosure, please include the initials of the people who were granted the funding, as well as a URL of the funding body.

SUPPLEMENTARY MATERIAL

REFERENCES

OBSERVATIONAL STUDIES

* Abstract: Please include the study design, population and setting, number of participants, years during which the study took place (enrollment and follow up), length of follow up, and main outcome measures.

* Please ensure that the study is reported according to the RECORD guideline (available from https://www.record-statement.org) and include the completed checklist as Supporting Information. Please add the following statement, or similar, to the Methods: "This study is reported as per the Reporting of Studies Conducted using Observational Routinely-Collected Data (RECORD) guideline (S1 Checklist)." When completing the checklist, please use section and paragraph numbers, rather than page numbers.

* For all observational studies, in the manuscript text, please indicate: (1) the specific hypotheses you intended to test, (2) the analytical methods by which you planned to test them, (3) the analyses you actually performed, and (4) when reported analyses differ from those that were planned, transparent explanations for differences that affect the reliability of the study's results. If a reported analysis was performed based on an interesting but unanticipated pattern in the data, please be clear that the analysis was data driven.

* Please state in the Methods section whether the study had a prospective protocol or analysis plan. If a prospective analysis plan (from your funding proposal, IRB or other ethics committee submission, study protocol, or other planning document written before analyzing the data) was used in designing the study, please include the relevant document(s) with your revised manuscript as a Supporting Information file to be published alongside your study and cite it in the Methods section. A legend for this file should be included at the end of your manuscript. If no such document exists, please make sure that the Methods section transparently describes when analyses were planned, and when/why any data-driven changes to analyses took place. Changes in the analysis, including those made in response to peer review comments, should be identified as such in the Methods section of the paper, with rationale.

---

## [Decision Letter · Decision Letter 2]

12 Dec 2025

Dear Dr. Deng,

Thank you very much for re-submitting your manuscript "The Role of Comorbidities in the Associations between Air Pollution and Alzheimer's Disease: A National Cohort Study in the American Medicare Population" (PMEDICINE-D-25-02362R2) for review by PLOS Medicine.

I have discussed the paper with my colleagues and the academic editor and it was also seen again by 2 reviewers. I am pleased to say that provided the remaining editorial and production issues are dealt with we are planning to accept the paper for publication in the journal after some minor concerns from the academic editor and the reviewers are addressed.

Specifically, the academic editor asked for some further discussion of the potential impact that AD misdiagnosis may have on the results, especially given that the increase in the risk for AD is rather small. Moreover, he raised concerns about the contribution of vascular mechanisms to the increased risk for AD in participants who suffered from stroke(s). One of the reviewers asked for for a further justification of the 5-year duration of the clean period.

The remaining issues that need to be addressed are listed in more detail at the end of this email. Any accompanying reviewer attachments can be seen via the link below. Please take these into account before resubmitting your manuscript:

[LINK]

We look forward to receiving the revised manuscript by Dec 12 2025 11:59PM.   

Sincerely,

Evangelia Fourli

Associate Editor 

PLOS Medicine

plosmedicine.org

Requests from Editors:

GENERAL EDITORIAL REQUESTS

* At this stage, we ask that you include a short, non-technical Author Summary of your research to make findings accessible to a wide audience that includes both scientists and non-scientists. The Author Summary should immediately follow the Abstract in your revised manuscript. This text is subject to editorial change and should be distinct from the scientific abstract. Ideally each sub-heading should contain 2-3 single sentence, concise bullet points containing the most salient points from your study. In the final bullet point of ‘What Do These Findings Mean?’ Please include the main limitations of the study in non-technical language.

Please see our author guidelines for more information: https://journals.plos.org/plosmedicine/s/revising-your-manuscript#loc-author-summary.

* Please confirm that your title complies with PLOS Medicine's style. Your title must be nondeclarative and not a question. It should begin with main concept if possible. "Effect of" should be used only if causality can be inferred, i.e., for an RCT. Please place the study design ("A randomized controlled trial," "A retrospective study," "A modelling study," etc.) in the subtitle (ie, after a colon).

* Please confirm that your abstract complies with our requirements, including format (three sections: Background, Methods and Findings, and Conclusions) and providing all the information relevant to this study type https://journals.plos.org/plosmedicine/s/submission-guidelines#loc-abstract

* Please ensure that the Introduction ends with a clear description of the study question or hypothesis.

* Please ensure that all abbreviations are defined at first use throughout the text.

* Please confirm that all numbers presented in the abstract are present and identical to numbers presented in the main manuscript text.

GENERAL

* Please review your text for claims of novelty or primacy (e.g. 'for the first time') and remove this language. In addition, please check that any use of statistical terms (such as trend or significant) are supported by the data, and if not please remove them.

* In the author summary, in the final bullet point of 'What Do These Findings Mean?', please include the main limitations of the study in non-technical language.

* Please provide the unadjusted comparisons as well as the adjusted comparisons in all relevant Tables

FUNDING STATEMENT

* The funding statement should include: specific grant numbers, initials of authors who received each award, URLs to sponsors’ websites. Also, please state whether any sponsors or funders (other than the named authors) played any role in study design, data collection and analysis, the decision to publish, or preparation of the manuscript. If they had no role in the research, include this sentence: “The funders had no role in study design, data collection and analysis, decision to publish, or preparation of the manuscript.”

COMPETING INTERESTS STATEMENT

* All authors must declare their relevant competing interests per the PLOS policy, which can be seen here: https://journals.plos.org/plosmedicine/s/competing-interests For authors with ties to industry, please indicate whether any of the interests has a financial stake in the results of the current study.

DATA AVAILABILITY

* PLOS Medicine requires that the de-identified data underlying the specific results in a published article be made available, without restrictions on access, in a public repository or as Supporting Information at the time of article publication, provided it is legal and ethical to do so. Please see the policy at

http://journals.plos.org/plosmedicine/s/data-availability

and FAQs at

http://journals.plos.org/plosmedicine/s/data-availability#loc-faqs-for-data-policy

* The Data Availability Statement (DAS) requires revision. For each data source used in your study:

OBSERVATIONAL, COHORT, CROSS-SECTIONAL, AND CASE CONTROL STUDIES

* Did your study have a prospective protocol or analysis plan? Please state this (either way) early in the Methods section.

* Your study is observational and therefore causality cannot be inferred. Please remove language that implies causality and refer to associations instead.

* For all observational studies, in the manuscript text, please indicate: (1) the specific hypotheses you intended to test, (2) the analytical methods by which you planned to test them, (3) the analyses you actually performed, and (4) when reported analyses differ from those that were planned, transparent explanations for differences that affect the reliability of the study's results. If a reported analysis was performed based on an interesting but unanticipated pattern in the data, please be clear that the analysis was data-driven.

Comments from the Academic Editor:

The Academic Editor expressed concerns about how the clinical diagnosis was conducted and how it may have impacted the results: Authors used the ICD-9/10 codes; however, a paper from Pippenger et al., 2001 that looked at "Neurologists' use of ICD-9 codes for dementia" concluded that "The specific code for AD, 331.0, was used for only 36.5% of patients judged by the neurologist to have AD as the most likely diagnosis. Other codes used were not inaccurate but would result in lower reimbursement". With this in mind and given the small increase in the risk for AD, this inaccuracy in diagnosis could be quite meaningful. Moreover, even when AD is a reasonable diagnosis, there is comorbid vascular or other pathology, which may make the difference to whether the patient presents with dementia or not. My particular query is about vascular pathology. The fact that stroke increased the risk makes me suspect that vascular mechanisms are possibly making a contribution. There is much evidence to suggest that cerebrovascular disease and AD are independent processes that synergistically contribute to cognitive impairment.

We ask the authors to address these concerns by the academic editor as well as the following requirement from reviewer #1.

Comments from Reviewers:

Reviewer #1: We thank the authors for addressing our previous concerns. The 5-year duration of the clean period before cohort entry (in the reply to Comment 5) might be further justified, if possible.

Reviewer #2: Great job!

[LINK]

---

## [Editor Report · Decision Letter 3]

13 Jan 2026

Dear Dr Deng, 

On behalf of my colleagues and the Academic Editor, Dr Sachdev, I am pleased to inform you that we have agreed to publish your manuscript "The Role of Comorbidities in the Associations between Air Pollution and Alzheimer's Disease: A National Cohort Study in the American Medicare Population" (PMEDICINE-D-25-02362R3) in PLOS Medicine.

PRESS

Sincerely, 

Evangelia Fourli 

Senior Editor 

PLOS Medicine